# Evaluation of the Potential of the Hard Clam (*Meretrix meretrix*) Shell Which Can Be Used as the Bioindicator for Heavy Metal Accumulation

Yen-Ting Lin [1,†], Fang-Chi Chang [1,†], Ming-Tsung Chung [2], Ze-Kai Li [1], Chang-Chuan Chan [3,4,*], Ying-Sheng Huang [5], Kuo-Fang Huang [6] and Yu-San Han [1,*]

1 Institute of Fisheries Science, College of Life Science, National Taiwan University, Taipei 10617, Taiwan
2 Atmosphere and Ocean Research Institute, The University of Tokyo, 5-1-5, Kashiwanoha, Kashiwa-shi 277-8564, Chiba, Japan
3 Department of Public Health, College of Public Health, National Taiwan University, Taipei 10055, Taiwan
4 Institute of Environmental and Occupational Health Sciences, College of Public Health, National Taiwan University, Taipei 10055, Taiwan
5 Fresh Water Research Center, Fisheries Research Institute, Council of Agriculture, Changhua 50562, Taiwan
6 Institute of Earth Sciences, Academia Sinica, 128, Sec. 2, Academia Road, Nangang, Taipei 11529, Taiwan
* Correspondence: ccchan@ntu.edu.tw (C.-C.C.); yshan@ntu.edu.tw (Y.-S.H.)
† These authors contributed equally to this work.

**Abstract:** Clams, *Meretrix meretrix*, were one of the favorite aquatic products in Taiwan and the world. It was reported that the water pollutants such as heavy metals and chemicals might accumulate in the clam body and shell through filter-feeding behavior. Thus, the bivalves could be used as bioindicators in the aquatic environment. The present study analyzed 20 trace elements, Mg, Sr, Li, Cd, Ba, Mn, Al, U, Ti, Pb, Nd, B, S, Zn, Fe, P, Na, K, Cu, and Ni, in the shells of the clams collected from the wild coastal area and cultured ponds to evaluate which elements have the potential to be biomarkers. The concentrations of 20 elements were detected by Solution-Based Inductively Coupled Plasma Mass Spectrometry (SB-ICPMS). Among them, Cd, Al, U, Ti, Nd, S, and K were below the detection limit. The remaining elements were analyzed for accumulation. The levels of Zn, Fe, and Ni possess high variation in an identical environment; so, a PCA was conducted without these three elements to reduce noise. The PCA result proved that the clam could absorb specific trace elements from the habitat. After a period of time, the contents of the absorption in the shells of the clams living in an identical environment became more similar. The analyzed element, Pb, was not found with a special difference in this study. The levels of Na, Sr, Mg, B, Mn, P, Ba, Li, and Cu reflected different sampling sites, which suggested that these elements in the species *M. meretrix* had the potential to be used as biomarkers for assessing heavy metal accumulation in the environment.

**Keywords:** bioindicator; environmental pollution; heavy metals; bivalves; mollusks; filter-feeding behavior





## 1. Introduction

Taiwan is an island state surrounded by sea, with a well-developed fishing and aquaculture industry. In particular, the mean annual production of hard clams in the past ten years has exceeded 55,000 tons [1]. The culture species of the hard clam is mostly *Meretrix meretrix*, which is one of the most commonly consumed mollusks in Taiwan [2]. *M. meretrix* belongs to Bivalvia in Mollusca, with a soft tissue body covered by two shells with symmetry. Bivalves, including mussels, oysters, and clams, are filter feeders and have been known to be able to accumulate organic and metallic pollutants from the water and sediment. They are frequently used as bioindicators in the aquatic environment [3–7]. Usually, the level of pollutant accumulated in the muscle tissue of mussels is used for assessing the level of pollution in their habitat.

Over the past several decades, industrialization and urban development have been taking place in Taiwan, accompanied by increasing organic and heavy metal pollution of the coastal environment, especially near estuaries [8–10]. Generally, such contaminants were transported from their sources through the river system and deposited downstream. The estuary is a potential sink for these pollutants for a long period of time and might lead to bioaccumulation in the food chain [11]. For instance, the discoloration of mariculture oysters, *Crassostrea gigas*, known as green oysters, occurring in the Erhjin Chi coastal waters in 1986, was caused by the high levels of copper ions released by nearby industries into the water [9,12–15]. On the other hand, the wild hard clam *M. meretrix* is commonly distributed on the sandy coast of western Taiwan and is mainly cultured in southwestern Taiwan (Changhua, Yunlin, Chiayi Counties, and Tainan City) [2,16]. The clam farms are usually located near coastal areas because saline water is needed for clam culturing. The extraction of contaminated water might bring such pollutants into the cultured pond and accumulate in the clam through their filter-feeding behavior.

The growth of clams is accompanied by the formation of growth rings recorded on their symmetry shell. A previous study reported that mineral composition and structure in the shells of bivalves might be affected by environmental change [17]. Due to the dominant component of the clam shells being calcium carbonate crystals, the trace elements would permanently accumulate in the carbonate form of the clamshell [18]. The heavy metal pollutants or elements not commonly existing at high levels in the organism would have a chance to accumulate in the shell depending on its body mechanism preference through tissue absorption [19]. According to this characteristic, shells have the potential to be useful indicators of environmental change and pollution and to provide a historical record throughout the organism's lifetime. Moreover, shells exhibit less variability than the living organism's tissue. For example, the concentration of Pb was evaluated by the shells of *Mytilus edulis* [20] and *Arctica islandica* [21]. The shell of *Trachycardium lacunosum* was used to monitor metals: Al, Zn, Fe, Mn, Ni, V, Co, Cr, and Cu in the marine environment [22]. The level of Cu in the shell of *Anodontites trapesialis* was sensitive to aquatic environments [23]. Different species of mollusks have different abilities to accumulate heavy metals [24].

In this study, we aimed to use Solution-Based Inductively Coupled Plasma Mass Spectrometry (SB-ICPMS) to determine the levels of 20 elements (Mg, Sr, Li, Cd, Ba, Mn, Al, U, Ti, Pb, Nd, B, S, Zn, Fe, P, Na, K, Cu, and Ni) in the shells of *M. meretrix* collected from the coastal area and cultured ponds of Taiwan. Then, we analyzed whether some elements had the potential to be biomarkers in monitoring pollution in the coastal environment.

## 2. Materials and Methods

### 2.1. Sample Collection

- Wild samples

Wild hard clams *M. meretrix* were captured from April to July 2019 from the Danshui River estuary in New Taipei City, the Siangshan Wetland in Hsinchu City, the coastal area of Xianxi in Changhua County, and the southern estuary of Zhuoshui River in Yunlin County (Figure 1, Table 1); all of these are the main production areas for wild *M. meretrix* in Taiwan.

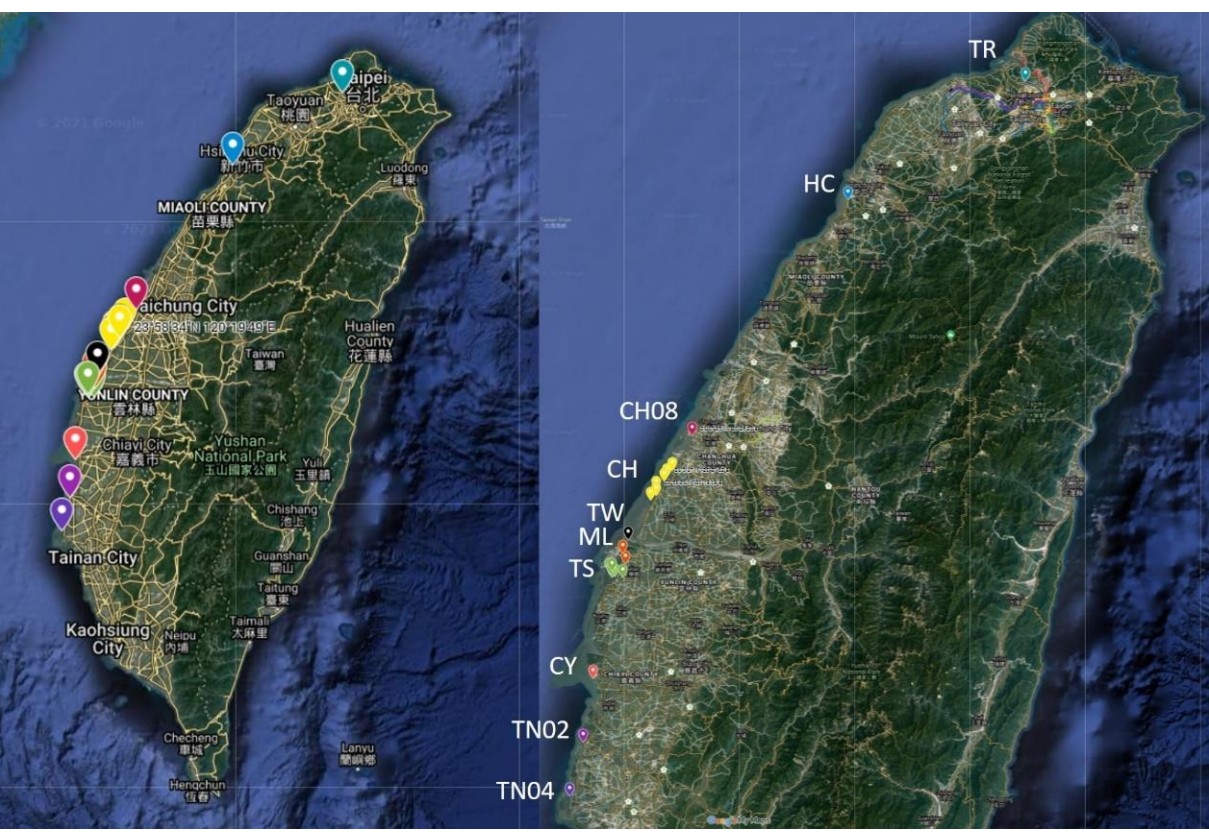

**Figure 1.** Sampling sites along the coastal area of Taiwan.

**Table 1.** The clam samples for analysis.

| Sample ID | Collected Region | Wild/Aquaculture |
|:---:|:---:|:---:|
| DS-w-1 | Danshui River estuary | wild |
| DS-w-2 | Danshui River estuary | wild |
| DS-w-3 | Danshui River estuary | wild |
| SS-w-1 | Siangshan Wetland (Hsinchu) | wild |
| SS-w-2 | Siangshan Wetland (Hsinchu) | wild |
| SS-w-3 | Siangshan Wetland (Hsinchu) | wild |
| XX-w-1 | coastal area of Xianxi | wild |
| XX-w-2 | coastal area of Xianxi | wild |
| XX-w-3 | coastal area of Xianxi | wild |
| ZS-w-1 | Zhuoshui River estuary | wild |
| ZS-w-2 | Zhuoshui River estuary | wild |
| ZS-w-3 | Zhuoshui River estuary | wild |
| CH01-1 | aquatic farms in Changhua | Aquaculture |
| CH01-2 | aquatic farms in Changhua | Aquaculture |
| CH01-3 | aquatic farms in Changhua | Aquaculture |
| CH02-1 | aquatic farms in Changhua | Aquaculture |
| CH02-2 | aquatic farms in Changhua | Aquaculture |
| CH02-3 | aquatic farms in Changhua | Aquaculture |
| CH03-1 | aquatic farms in Changhua | Aquaculture |
| CH03-2 | aquatic farms in Changhua | Aquaculture |
| CH03-3 | aquatic farms in Changhua | Aquaculture |
| CH04-1 | aquatic farms in Changhua | Aquaculture |
| CH04-2 | aquatic farms in Changhua | Aquaculture |
| CH04-3 | aquatic farms in Changhua | Aquaculture |
| CH05-1 | aquatic farms in Changhua | Aquaculture |
| CH05-2 | aquatic farms in Changhua | Aquaculture |

**Table 1.** *Cont.*

| Sample ID | Collected Region | Wild/Aquaculture |
|---|---|---|
| CH05-3 | aquatic farms in Changhua | Aquaculture |
| CH06-1 | aquatic farms in Changhua | Aquaculture |
| CH06-2 | aquatic farms in Changhua | Aquaculture |
| CH06-3 | aquatic farms in Changhua | Aquaculture |
| ML01-1 | aquatic farms in Mailiao (Yunlin) | Aquaculture |
| ML01-2 | aquatic farms in Mailiao (Yunlin) | Aquaculture |
| ML01-3 | aquatic farms in Mailiao (Yunlin) | Aquaculture |
| ML02-1 | aquatic farms in Mailiao (Yunlin) | Aquaculture |
| ML02-2 | aquatic farms in Mailiao (Yunlin) | Aquaculture |
| ML02-3 | aquatic farms in Mailiao (Yunlin) | Aquaculture |
| TX01-1 | aquatic farms in Taixi (Yunlin) | Aquaculture |
| TX01-2 | aquatic farms in Taixi (Yunlin) | Aquaculture |
| TX01-3 | aquatic farms in Taixi (Yunlin) | Aquaculture |
| TX02-1 | aquatic farms in Taixi (Yunlin) | Aquaculture |
| TX02-2 | aquatic farms in Taixi (Yunlin) | Aquaculture |
| TX02-3 | aquatic farms in Taixi (Yunlin) | Aquaculture |
| TX03-1 | aquatic farms in Taixi (Yunlin) | Aquaculture |
| TX03-2 | aquatic farms in Taixi (Yunlin) | Aquaculture |
| TX03-3 | aquatic farms in Taixi (Yunlin) | Aquaculture |
| TX04-1 | aquatic farms in Taixi (Yunlin) | Aquaculture |
| TX04-2 | aquatic farms in Taixi (Yunlin) | Aquaculture |
| TX04-3 | aquatic farms in Taixi (Yunlin) | Aquaculture |
| TX05-1 | aquatic farms in Taixi (Yunlin) | Aquaculture |
| TX05-2 | aquatic farms in Taixi (Yunlin) | Aquaculture |
| TX05-3 | aquatic farms in Taixi (Yunlin) | Aquaculture |
| TX06-1 | aquatic farms in Taixi (Yunlin) | Aquaculture |
| TX06-2 | aquatic farms in Taixi (Yunlin) | Aquaculture |
| TX06-3 | aquatic farms in Taixi (Yunlin) | Aquaculture |
| CY-1 | aquatic farms in Chiayi | Aquaculture |
| CY-2 | aquatic farms in Chiayi | Aquaculture |
| CY-3 | aquatic farms in Chiayi | Aquaculture |
| BM-1 | aquatic farms in Beimen (Tainan) | Aquaculture |
| BM-2 | aquatic farms in Beimen (Tainan) | Aquaculture |
| BM-3 | aquatic farms in Beimen (Tainan) | Aquaculture |
| QG-1 | aquatic farms in Qigu (Tainan) | Aquaculture |
| QG-2 | aquatic farms in Qigu (Tainan) | Aquaculture |
| QG-3 | aquatic farms in Qigu (Tainan) | Aquaculture |

- Cultured samples

Cultured *M. meretrix* were collected from 17 aquatic farms in the Changhua, Yunlin, Chiayi counties and Tainan City from April to August 2019 (Figure 1, Table 1). These places were the primary source of cultured hard clams in Taiwan.

After collection, the hard clams were transferred to the lab of National Taiwan University and sacrificed immediately after arriving. The entire soft tissue of the clam was separated for DNA species identification to make sure the samples were all *M. meretrix*.

### 2.2. Trace Elements Analysis

A physical grinding removed the organic layer on the surface of the shell to reduce the organic matter (Figure 2) in the channel of the Solution-Based Inductively Coupled Plasma Mass Spectrometry (SB-ICPMS), which may cause machine damage. The outer, middle, and inner sites of every shell sample were scraped off, and the powders were collected and weighted. The outer site showed the accumulation of elements in the shell when the clams were captured. The middle and inner sites represented the elements where the clams grew up and the larva stage, respectively. Four hundred microliters of sodium hypochlorite

was added into every powder sample to remove the organic matter, and the samples were processed many times by centrifugation and washing with MQ-water. Then, 450 μL of 0.001 N nitric acid was added to erode the surface of powder particles, followed by centrifugation. In the last step, 450 μL of 0.1 N nitric acid was added to dissolve the powder and every solution sample was diluted to an equal concentration for SB-ICPMS detection.

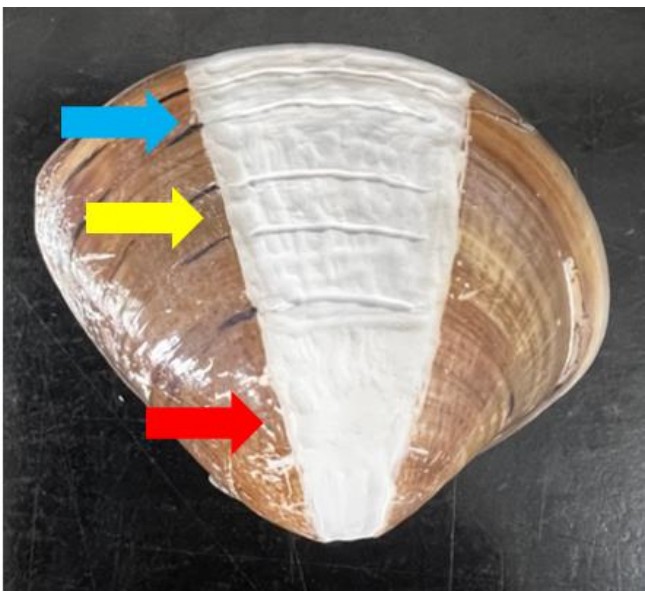

**Figure 2.** The shell sample of *M. meretrix* after grinding for removal of the surface to reduce organic matter. The area in the red arrow was the inner site; the yellow arrow was referred to as the middle site; and the blue arrow was mentioned as the outer site.

An SB-ICPMS was used to analyze the element concentration of the samples. This study tested 20 elements: Mg, Sr, Li, Cd, Ba, Mn, Al, U, Ti, Pb, Nd, B, S, Zn, Fe, P, Na, K, Cu, and Ni. Calibration solutions containing known concentrations of each element were made, and calibration curves were drawn for each element. Ca was used as a reference because it is the major constitutional element of the shell.

*2.3. Statistical Analysis*

All the results were analyzed by SPSS 24.0 for Windows (IBM SPSS Statistics 24.0), and the data were expressed as mean ± standard deviation (SD). The Kruskal–Wallis test was used for one-way analysis of variance. If there was a significant difference, Dunn's test was used to test between the groups to confirm whether there was a difference in means between the groups. Significant difference was considered at $p < 0.05$.

For an overview of the relative concentration levels of elements and differences between samples, the multivariate analysis called principal component analysis (PCA) was performed on the data. The PCA method seeks out vectors called main components. The model could be described in fewer principal components than the original number of variables without losing a considerable amount of information. The PCA was performed by XLSTAT software to evaluate what the elements in the shell might show about the environmental variation where the M. meretrix lived.

**3. Results**

*3.1. Trace Elements Concentration in the Shell of M. meretrix*

The specimens from all the sampling sites were identified as *M. Meretrix*. Each cultured pond or wild location was randomly selected for three individuals for analysis. The measured trace elements concentration in the shell of *M. meretrix* was listed in Table S1. Among them, the elements Cd, Al, U, Ti, Nd, S, and K were not shown because these

elements were below the detection limit. The sample ML02-3-I was missing because the concentration was too low after sample processing to proceed with the analysis. To compare the accumulation levels of the trace elements in the life stages, the inner, middle, and outer sites of the ring on the shell represent the early, middle, and present stages, respectively.

*3.2. Inner Site of the Shells*

According to Figure 2, the gray bar represents the average concentration (ppm) of the trace elements on the inner side of the shells. A high variation of accumulation levels showed in Zn and Ni, and there was no significant difference in every sample ($p > 0.05$). For the level of Na, there was no significant difference in the wild samples ($p > 0.05$); BM had almost the highest level in the cultured samples ($p < 0.05$). The level of Sr, SS-w was higher than XX-w and ZS-w in the wild samples ($p < 0.05$); QG and CY were higher than BM in the cultured samples ($p < 0.05$). For the level of Mg, there was no significant difference in the wild samples ($p > 0.05$); CY was higher than BM in the cultured samples ($p < 0.05$). The levels of Fe and SS-w had the highest level in the wild samples ($p < 0.05$); there was no significant difference in the cultured samples ($p > 0.05$). The level of B, ZS-w had the lowest level in the wild samples ($p < 0.05$); QG had the highest level in the cultured samples ($p < 0.05$). For the level of Mn, there was no significant difference in the wild samples ($p > 0.05$); CY had the highest level in the cultured samples ($p < 0.05$). For the level of P, there was no significant difference in the wild samples ($p > 0.05$); there was also no difference in the cultured samples ($p > 0.05$). However, many elements in the reared samples were higher than in the wild samples. Regarding the level of Ba, there was no significant difference in the wild samples ($p > 0.05$); TX had the highest level in cultured samples ($p < 0.05$). Many cultured samples were higher than the wild samples. Regarding the level of Li, there was no significant difference in the wild samples ($p > 0.05$); there was also no difference in the cultured samples ($p > 0.05$). For the level of Cu, there was no significant difference in the wild samples ($p > 0.05$); there was also no difference in the cultured samples ($p > 0.05$). The higher average concentration showed a higher variation, and the cultured samples with a high concentration of Cu were higher than most of the wild samples ($p < 0.05$). Regarding the level of Pb, there was no significant difference in every sample ($p > 0.05$).

*3.3. Middle Site of the Shells*

According to Figure 3, the orange bar represents the average concentration (ppm) of the trace elements in the middle site of the shells. The high variation of the accumulation level was found in Zn and Ni, and there was no significant difference in every sample ($p > 0.05$). Regarding the level of Na, there was no significant difference in the wild samples ($p > 0.05$); TX was significantly higher than CH, CY, and QG in the cultured samples ($p < 0.05$). The wild samples except for XX-w were higher than most of the cultured samples. The level of Sr, SS-w had the highest level in the wild samples ($p < 0.05$); there was no significant difference in the cultured samples ($p > 0.05$). The level of Mg, ZS-w was significantly higher than SS-w in the wild samples ($p < 0.05$); there was no significant difference in the cultured samples ($p > 0.05$).

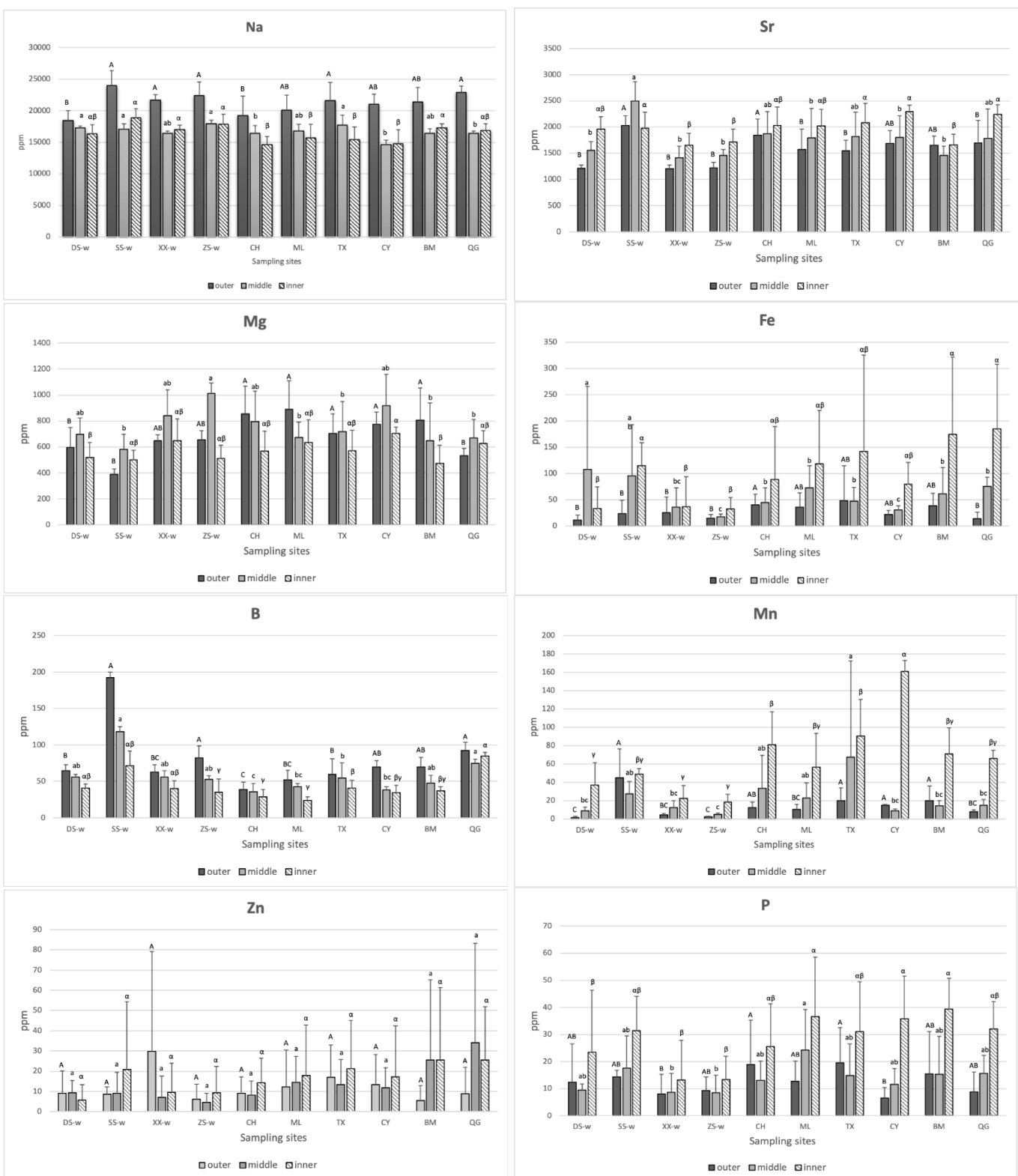

**Figure 3.** *Cont.*

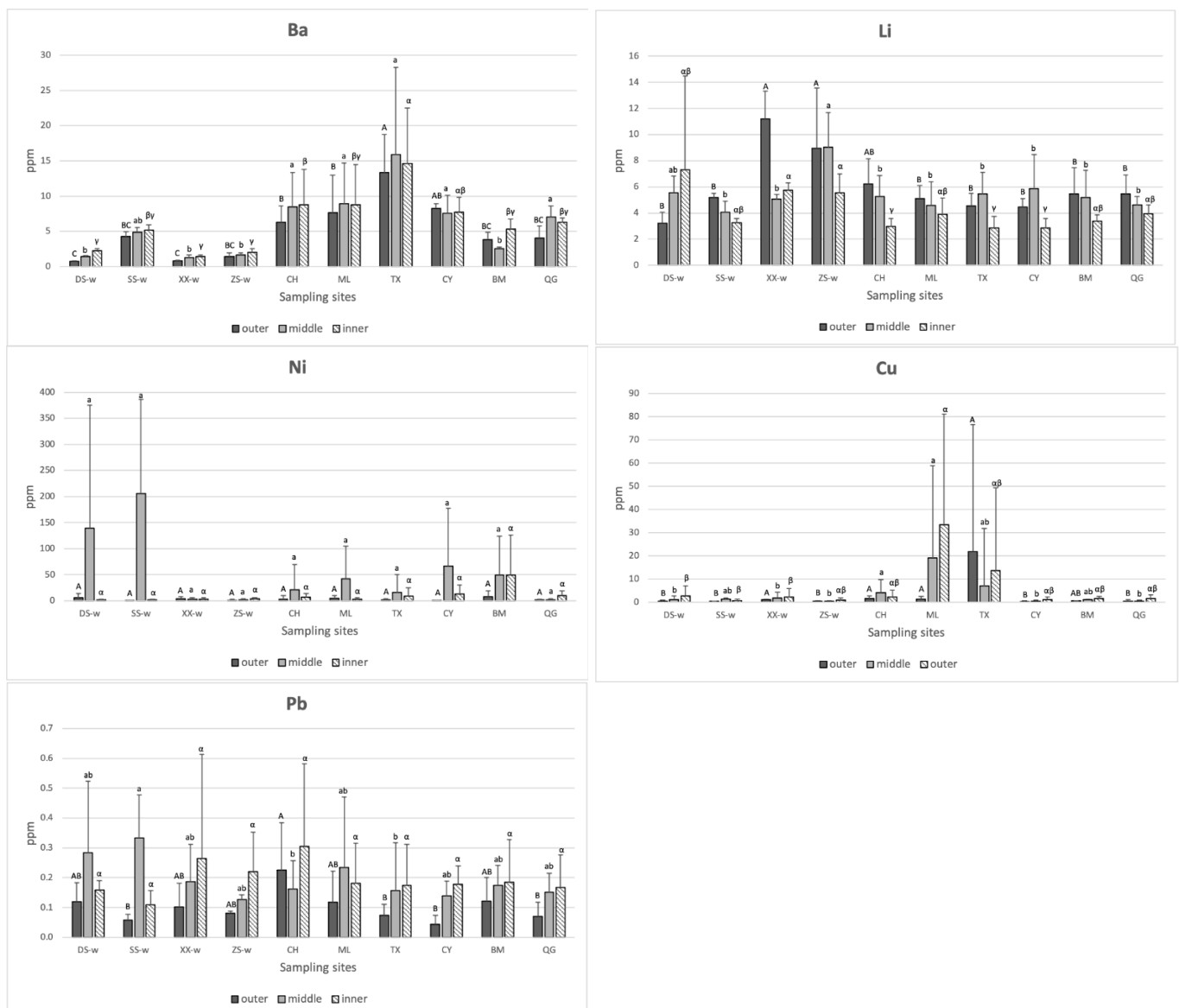

**Figure 3.** The 13 trace elements in the outer, middle, and inner site of the shells. All the results were represented as mean ± standard deviation (SD). Different capital letters among the groups indicate that an element differed significantly ($p < 0.05$) in the outer; different lower case letters among the groups indicate that an element differed significantly ($p < 0.05$) in the middle; different Greek alphabet letters among the groups indicate that an element differed significantly ($p < 0.05$) in the inner.

In the wild samples, DS-w had higher levels of Fe than XX-w and ZS-w in the wild samples ($p < 0.05$); SS-w had higher levels of Fe than ZS-w ($p < 0.05$). In the cultured samples, CY showed the lowest concentration of Fe ($p < 0.05$). Regarding the level of B, there was no significant difference in the wild samples ($p > 0.05$); TX, BM, and QG were significantly higher than CH in the cultured samples ($p < 0.05$). The level of Mn, SS-w was significantly higher than ZS-w in the wild samples ($p < 0.05$); TX was significantly higher than CY, BM, and QG in the cultured samples ($p < 0.05$). Regarding the level of P, there was no significant difference in the wild samples ($p > 0.05$); there was also no difference in the cultured samples ($p > 0.05$). Regarding the level of Ba, there was no significant difference in the wild samples ($p > 0.05$); BM was the lowest level in the cultured samples ($p < 0.05$). Interestingly, the cultured samples were higher than the wild samples except for BM ($p < 0.05$). The level of Li, ZS-w was higher than XX-w and SS-w in the wild samples

($p < 0.05$); there was no significant difference in the cultured samples ($p > 0.05$). Regarding the level of Cu, there was no significant difference in the wild samples ($p > 0.05$); CH and ML were higher than CY and QG in the cultured samples ($p < 0.05$). Furthermore, the higher average concentration showed a higher variation, and the cultured samples with a high concentration of Cu were higher than most of the wild samples ($p < 0.05$). Regarding the level of Pb, there was no significant difference in the wild samples ($p > 0.05$); CH was higher than TX, CY, and QG in the cultured samples ($p < 0.05$).

### 3.4. Outer Site of the Shells

The outer site of the shells revealed the recent environments in which the animals had lived. According to Figure 3, the blue bar represents the average concentration (ppm) of the trace elements in the outer site of the shells. A high variation of accumulation level was found in Zn and Ni, and there was no significant difference in every sample ($p > 0.05$). The level of Na, DS-w had the lowest level in the wild samples ($p < 0.05$); TX and QG were significantly higher than CH in the cultured samples. The level of Sr, SS-w had the highest level in the wild samples ($p < 0.05$); CH was significantly higher than ML, TX, and QG in the cultured samples ($p < 0.05$). Regarding the level of Mg, there was no significant difference in the wild samples ($p > 0.05$); QG was significantly lower than the other cultured samples ($p < 0.05$). Interestingly, the samples from the north of Taiwan such as DS-w and SS-w were shown to be significantly lower than all the cultured samples except for QG ($p < 0.05$). Regarding the level of Fe, there was no significant difference in the wild samples ($p > 0.05$); CH was significantly higher than QG in the cultured samples ($p < 0.05$). The levels of B, SS-w and ZS-w had the higher level in the wild samples ($p < 0.05$). In the cultured samples, QG was significantly higher than CH, ML, and TX ($p < 0.05$). Among them, TX was significantly higher than CH ($p < 0.05$). The level of Mn, SS-w had the highest level in the wild samples ($p < 0.05$); TX and BM were significantly higher than ML and QG in the cultured samples ($p < 0.05$). Regarding the level of P, there was no significant difference in the wild samples ($p > 0.05$); CH and TX were significantly higher than CY in the cultured samples ($p < 0.05$). Regarding the level of Ba, there was no significant difference in the wild samples ($p > 0.05$); TX showed the highest level in the cultured samples ($p < 0.05$) but had no significant difference with CY. The levels of Li, XX-w and ZS-w had the higher levels in the wild samples ($p < 0.05$); there was no significant difference in the cultured samples ($p > 0.05$). XX-w and ZS-w showed the highest level in all samples ($p < 0.05$) but had no significant difference with CH. The level of Cu, XX-w had the highest level in the wild samples ($p < 0.05$); CH, ML, and TX were significantly higher than CY and QG in the cultured samples ($p < 0.05$). Furthermore, the higher average concentration showed higher variation. Regarding the level of Pb, there was no significant difference in the wild samples ($p > 0.05$); CH was significantly higher than TX, CY, and QG in the cultured samples ($p < 0.05$).

### 3.5. Trace Elements in Three Stages of the M. meretrix

The data of the wild and cultured samples were combined for stage analysis, and the comparison of every trace element in the three stages is illustrated in Figure 4. Regarding the level of Na and B, the outer site was the highest and the inner site was the lowest in the three stages ($p < 0.05$), while the inner site was the highest and the outer site was the lowest ($p < 0.05$) in the level of Sr, Fe, and Mn. Regarding the level of Mg and Li, the inner site had a lower concentration than the outer and middle site ($p < 0.05$). Regarding the level of P, the outer site had a higher concentration than the middle and inner site ($p < 0.05$), while there was an opposite state in the level of Ni and Pb. There was no significant difference in the three stages with regard to the level of Zn, Ba, and Cu ($p > 0.05$).

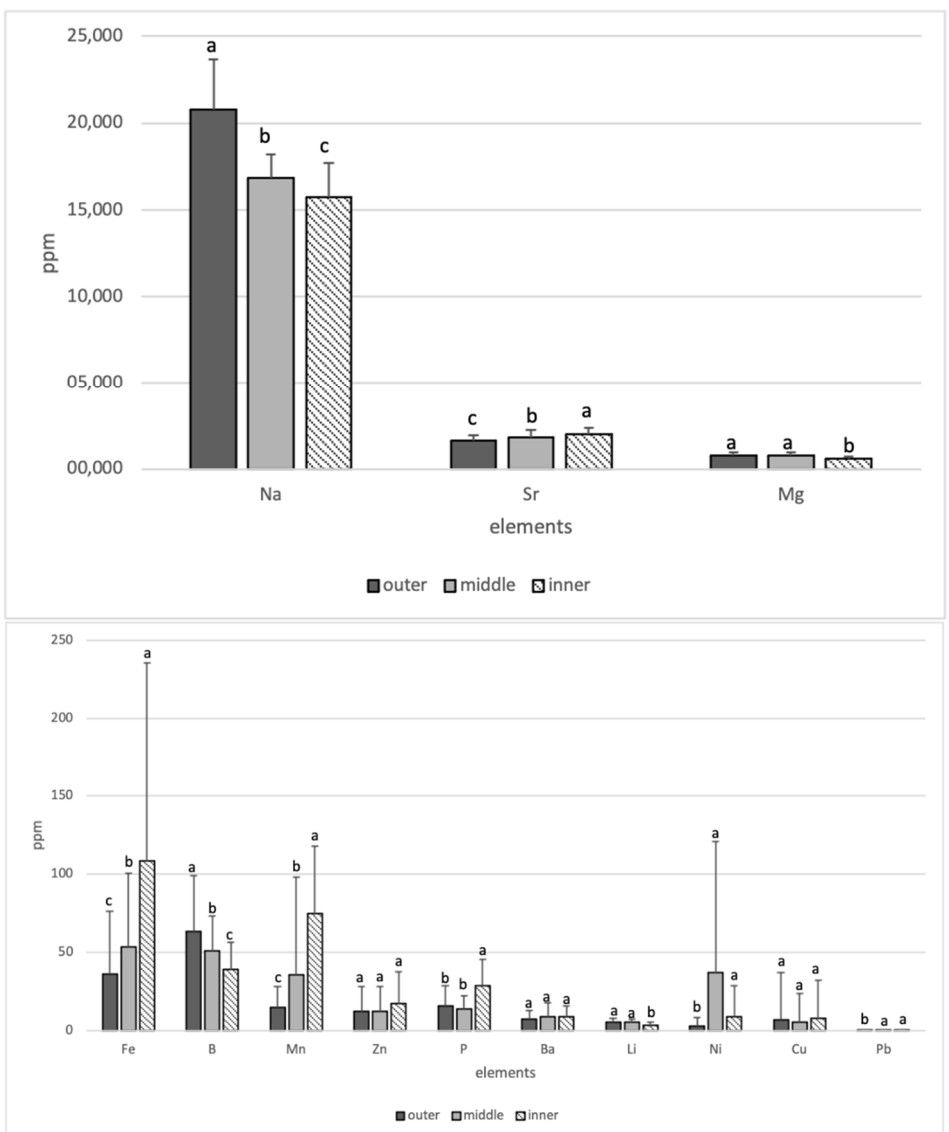

**Figure 4.** The comparison of the level of every trace element in three stages. All the results were represented in the form of mean ± standard deviation (SD). Different lower case letters among the groups indicate that an element differed significantly ($p < 0.05$) in middle.

*3.6. PCA Analysis of Trace Elements in the Three Sites of the Shell*

The data of CH01-1-I, CH04-1-I, CH04-3-O, ML02-3-O, TX02-1-M, TX03-1-M, and TX06-3-M were excluded because of detection error. Because of the high variation between the individuals collected from the identical sampling site, the Zn, Fe, and Ni were not properly used as indicators. These three elements were excluded in principal component analysis (PCA) to reduce the noise. The data were processed by statistical techniques, and the PCA results were obtained. In the inner site of the shells, the principal components, in PC1, were Sr, Li, Ba, Mn, and Na; in PC2, it was B; and in PC3, it was Mg. The PCA result showed wild samples from the same location clustered together, except for the DS-w samples. The cultured samples from the identical cultured pond were also closer to each other. However, the Changhua and Taixi (Yunlin) cultured samples were not distinctly separated (Figure 5). Interestingly, the wild and cultured samples were obviously separated.

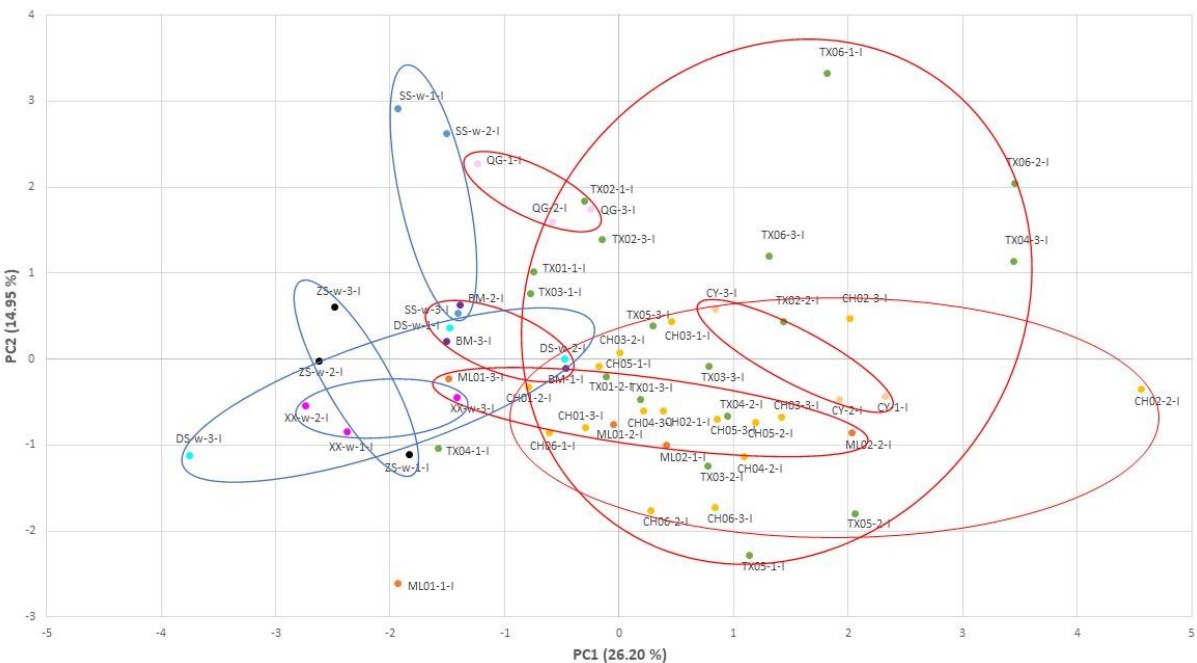

**Figure 5.** Principal component analysis (PCA) in the inner site of the shells: PCA plotted against the PC1 vs. PC2 axes. Different color of points in the figure means different locations.

In the middle site of the shells, the principal components, in PC1, were Mg and Li; in PC2, they were Sr, Ba, and Mn; and in PC3, they were Pb and P. The PCA result showed that the wild samples, which were considered to be staying in one place without migration during their growth, were mostly closer to each other when collected from the same location, while the cultured samples, especially those from Changhua and Taixi (Yunlin), were embedded together (Figure 6).

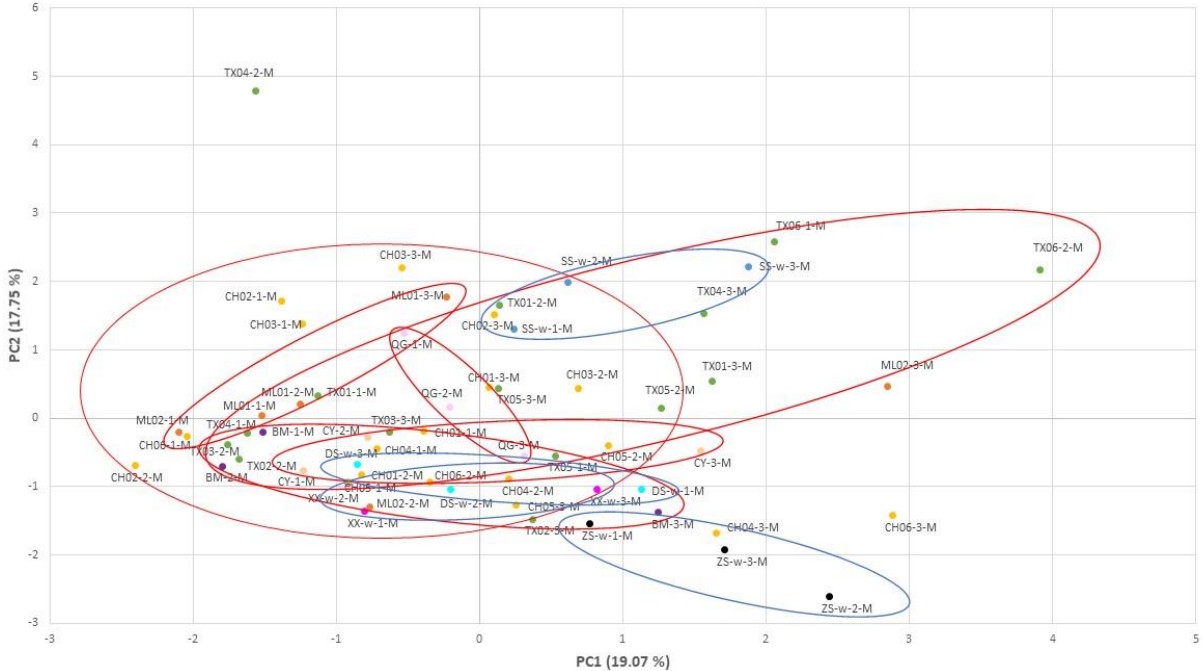

**Figure 6.** Principal component analysis (PCA) in the middle site of the shells: PCA plotted against the PC1 vs. PC2 axes. Different color of points in the figure means different locations.

In the outer site of the shells, the principal components, PC1, were Sr, Mn, and Na; in PC2, they were Mg, Pb, B, and P; in PC3, they were Li, Ba, and Cu. The PCA result showed that the individuals derived from the same location or cultured ponds were mostly close to each other, and the accumulation levels in the shells of the clams were closer as well (Figure 7). The results indicated that the accumulation of some trace elements would represent the environment where the samples were collected.

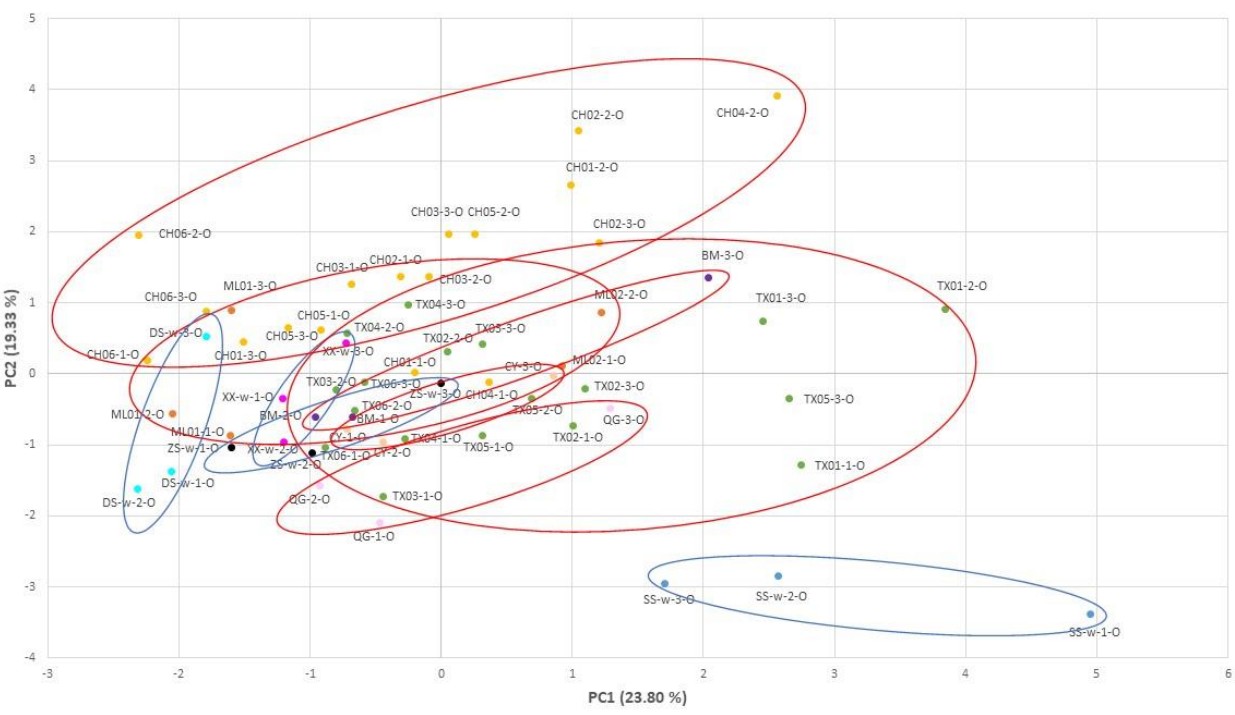

**Figure 7.** Principal component analysis (PCA) in the outer site of the shells: PCA plotted against the PC1 vs. PC2 axes. Different color of points in the figure means different locations.

## 4. Discussion

It was found that the tissues and the shell of *M. meretrix* had the potential to be used as bioindicators for the contamination of Zn [3,4,6,25] and Ni [26] in the water and sediment of an estuarine environment. On the other hand, Rashid et al. suggested that the ability of *M. meretrix* to regulate Zn may reduce its effectiveness in being used as a bio-monitoring organism for Zn [27]. In this study, a high variation of the accumulation level of Zn and Ni was found in all of the stages of the clam shells derived from all the sampling sites. This phenomenon suggested that the concentrations of these two elements may not be good candidates for bioindicators. On the other hand, according to the results of this research, there are five elements (Na, B, Sr, Fe, and Mn) that change the rate of accumulation between different growth stages of *M. meretrix*. Among them, Na and B showed that accumulation would increase along with individual development. On the other hand, the concentration of Sr, Fe, and Mn in the shell of *M. meretrix* would decrease along with growth.

The level of Na was usually related to the salinity of the water, where *M. meretrix* usually lived in a middle to high salinity environment. According to the data from the water monitoring, the salinity of the water used in the cultured ponds in Tainan City was the highest among all the investigation sites [2]. The levels of Na in the Tainan samples (BM and QG) were higher than those of the other cultured samples. The PCA results also showed that the cultured samples from the Tainan samples were closer to the wild group, which lives in the seashore with usually higher salinity. The water salinity in the cultured ponds of Changhua was the lowest among all the sampling sites [2], which coordinated with the lowest Na level shown in the Changhua samples. Boron is a mineral used for various purposes in the glass, ceramics, automotive, and paint industries [28]. In our study,

the wild samples from SS-w had the highest level of B, and it was reported that paint poured into the river near the Siangshan Wetland in September 2019 [29]. According to this report, the pouring of paint probably continued for a period of pollution time because samples were also detected with higher levels of B at the larva stage in SS-w. Furthermore, the variation of the boron concentration was very low, which implied that the shell of *M. meretrix* may be a good indicator for boron to reflect the real environment between the different growing stages. According to the previous study, Na and B show an increasing accumulating trend in ocean acidification and also present similar results in *Dicathais orbita* [30,31]. However, in the research on *Ruditapes philippinarum* a lower Na level in the shell of the offspring from an acidic exposure environment than that growing under a standard condition is shown. Still, both also present an increasing accumulation rate of Na along with the growing [32]. Ocean acidification may be the reason for the accumulation of Na, and B showed a decreasing trend in *M. meretrix*.

Strontium is usually associated with human activities, including factory manufacturing wastewater and household waste. In our results, Sr showed the highest level in all the groups of *M. meretrix* at the larval stage, but subsequently, it offered a decreasing tendency along with growth. The highest Sr level was detected in both the cultured and wild samples from the Siangshan Wetland in the Hsinchu region, which was recorded as having heavy metal pollution from nearby manufacturing factories. According to a previous study, the compound of strontium was usually used for the production of ceramics, glass, fireworks, fluorescent lamps, and drugs [33]; this is the same as the factory nearby the Hsinchu region.

In this study, the high variation of the accumulation level of iron was found in all of the stages of the clam shells from every sampling site. However, the results show the same tendency: the Fe level in the shell decreased along with the growth stage. This suggested that the concentrations of the Fe element in the shells were not that stable, and it may therefore not be a good biomarker. The source of Fe was related to industrial or mixed sources [34]. Both the cultured samples and the wild samples from SS-w were detected with one of the highest Fe concentrations, which meant that accumulation of Fe may be associated with human activities, which is nearly the same as with Sr. Moreover, in previous research, it was shown that the shell of *M. lusoria* could reflect a high level of Fe in its habitat during its different growing stages [35].

Another element that decreased with the growing of *M. meretrix* was manganese. The wild samples from SS-w were detected with the significantly highest concentration of Mn among all of the groups. As mentioned before, there are lots of industry areas and factories near the Hsinchu area. In a recent study, Mn was usually associated with the production of fireworks, glass, drugs, leather, and the manufacture of steel and iron [36]. Such human activities or waste emissions may cause the higher Mg levels in SS-w. As for the Hsinchu area, the Hsinchu Science Industrial Park (HSIP) is located in the upper stream of the Siangshan Wetland. The main composition of the industries in the HSIP included semiconductor industries and glass processing, and some parts were for the bio-technology and drag companies. From the previous research mentioned above, those factories may be the main emitters of Sr, Fe and Mn, which caused the highest concentration in SS-w.

Phosphorus usually exists in a phosphate form in the waters and is also an essential nutrient for plant growth. However, when excessive phosphate occurs in the waters, the algae tend to grow and reproduce numerously. After the algae break down, they consume a large amount of oxygen due to their corruption and decomposition, resulting in the mass killing of the aquatic creature [37]. In our results, many cultured samples had higher P levels than those of the wild samples. The source of organic phosphorus in waters usually came from feed and feces [38]. The cultured clams were usually cultivated with other fish (such as: *Chanos chanos*, *Liza macrolepis*); the waste from this co-cultivation of fish provided a rich organic phosphorus source [39]. As for the wild clams, which usually lived in the sandy sediment, the tide took away the organic matter, causing less P concentration, as shown by the wild samples XX-w and ZS-w in this study.

Some elements (P, Mg, Ba, Li, Cu, and Pb) do not show a significant tendency between the accumulation rate and the growth stage. The respiration rate of hard clams would decrease in seawater without magnesium ions [40]. In addition, the artificial seawater which was used in the culture pond was lacking in Mg, which would significantly reduce the digging ability of hard clams [41]. The lower level of Mg was found in the inner site of all the samples in our study, representing the clam's larva phase. The lower absorption and digging ability due to the low level of Mg in the larval clam coincided with its habit, which was to stay on the surface of the sand.

The industrial uses of Ba were wide and variable, and it was used to produce coating materials, brick, cement, glass, and rubber [42]. Most wild samples seemed to have low levels of Ba, except the samples from SS-w; in both the wild and the cultured samples from SS-w a higher concentration of Ba was detected, which suggested that Ba may be associated with the pollution caused by HSIP, as mentioned before.

In a study by Stoffyn-Egli, it was pointed out that Lithium showed the highest concentration in estuaries, which corresponds to our results [43], where the wild samples from the Xianxi River (XX-w) and the Zhuoshui River (ZS-w) both show a significantly higher level of Li than the others.

As for the level of copper, it showed non-regular results in our study, where it showed a high variation between different growth stages and differed in each sample site. The most probable reason for the cause of this may be the treatment which was conducted in the cultured pond or the pollution from the factory. As for the cultured samples, which mainly showed a higher Cu level, it might be because of the copper sulfate commonly used for sterilization in aquaculture [44]. When copper sulfate is added into the aquaculture system, it will dissociate to $Cu^{+2}$ and $SO_4^{-2}$, which is the Cu ion that was the primary accumulated type of copper for the clam.

Lead has been a great concern due to its harmful effects on human health and the environment. Pb may impair the normal neurological development of children even at low exposure levels and increase the risk of cardiovascular diseases and renal deficiencies in adults [45]. The accumulation of Pb was recorded in the bivalves' shells [21,46,47]. This study showed that all the sampling sites had no contamination with lead, and the national standard of lead pollution is 0.14 ppm [48].

It has been pointed out that the element composition of clamshells could be influenced by environmental factors such as water pollution [4], ocean acidification [30,31], and climate change [49]. Moreover, the excessive concentration of some elements may cause a health hazard for the environment or humans [50,51]. The results in this study showed a different accumulation pattern of the elements in the shell of *M. meretrix*, which were similar to those in the tissue or shell of another hard-shell clam [4,30,49]. In the results of Na, B, Sr, Fe, and Mn, there was a significant difference ($p < 0.05$) between different areas and times, which could indicate that *M. meretrix* has good potential as the biomarker of those elements. However, without the data of the water and sediment in the sampling area of this study, we could only find out the potential of using *M. meretrix* as a biomarker of specific elements.

The PCA results showed that the wild samples from the same sampling site, except for DS-w, would cluster together at the larva stage. This suggested that the same cohorts usually possess similar environments and thus have a similar element composition. It was reported that some of the larvae of the DS-w samples, which came from spawning farms all over Taiwan, might be released by the Fisheries Research Institute to maintain the clam biomass [52], (https://www.tfrin.gov.tw/News_Content.aspx?n=4090&s=230848 (accessed on 16 November 2019)); this might explain why there is a higher variation of element composition in the DS-w samples. In Taiwan, the aquaculture of the clam was usually divided into two stages, spawning and cultivation. The larvae for cultivation were usually obtained from the same spawning stations. The cultivation environment of the central region of Taiwan, such as in Changhua and Yunlin, was similar, which corresponded to the result in the inner shell (Figure 4). After the clam farmers purchased the juvenile clam, they transported it to their aquaculture ponds all over Taiwan; since then, the accumulation

in the shell of M. meretrix had been influenced by the different rearing environments, according to the results of the middle and outer parts of the shell, which also show the same separated tendency as other areas in cultured clams (Figures 5 and 6). Although cultured in different regions, the clam comes from almost the same parents; so, this may explain why the composition of elements in the shells of the cultured samples collected from each region could not wholly be separated.

Taken together, the hard clam *M. meretrix* was a wild distributed bivalve on the coast of Taiwan. Its shell could be used as a good bioindicator to show the environment change for the elements such as Na, Sr, Mg, B, Mn, P, Ba, Li, and Cu. Moreover, with the phenomenon of the element accumulation of Na, Sr, Fe, B, and Mn, which differ along with the growing stage, the use of them as biomarkers should be conducted carefully.

## 5. Conclusions

In conclusion, the shell of the hard clam *M. meretrix,* which is a wild distributed bivalve on the coast of Taiwan, could be used as an excellent biomarker to show the environment change for the elements such as Na, Sr, Mg, B, Mn, P, Ba, Li, and Cu. Moreover, since Na and B showed an increasing accumulation along with the development and Sr, Fe, and Mn, with a decreasing trend along with growth, detecting those elements should be handled more carefully. However, due to the lack of heavy metal accumulation data of the water and sediment in the sampling site of this research, one should be more cautious about using *M. meretrix* as the bioindicator. *M. meretrix* presented an excellent potential for serving as a bioindicator for heavy metal element accumulation on the coast, and for some specific elements with different accumulation patterns, and further studies should clarify the correlation between the patterns and the habitat.

**Supplementary Materials:** The following supporting information can be downloaded at: https://www.mdpi.com/article/10.3390/fishes7050290/s1, Tables S1–S4: the level of 13 trace elements in the outer, middle, and inner site of the shell.

**Author Contributions:** Y.-T.L. and F.-C.C. mainly conducted the experiments, analyzed the results, and wrote the manuscript. Data collections were performed by Z.-K.L.; Y.-S.H. (Yu-San Han) designed and supervised the experiments; M.-T.C., C.-C.C., Y.-S.H. (Ying-Sheng Huang), and K.-F.H. supervised the experiments. All authors have read and agreed to the published version of the manuscript.

**Funding:** The authors thank the Council of Agriculture, Executive Yuan, Taiwan, for funding this project (1082101011904-010201 m3, 108AS-1.2.1-ST-m3, and 109AS-1.2.2-ST-m1) and the National Science and Technology Council, Executive Yuan, Taiwan (NSTC 111-2313-B-002-016-MY3).

**Institutional Review Board Statement:** The study was approved by the Institutional Animal Care and Use Committee (IACUC), National Taiwan University (approval code NTU107-EL-00059, approved from 1 August 2018 to 31 July 2021).

**Data Availability Statement:** The data that support the findings of this study are available from the corresponding author, Yu-San Han, upon reasonable request.

**Conflicts of Interest:** The authors declare that they have no conflict of interests.

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
