# Peer review of "Evaluation of the Potential of the Hard Clam (Meretrix meretrix) Shell Which Can Be Used as the Bioindicator for Heavy Metal Accumulation"

_fishes, doi:10.3390/fishes7050290_

Round 1

Reviewer 1 Report

All corrections and questions should be done and answered before the start of publication process

LN/34---add bivalves , Mollusks and filter feeding behavior to the keywords

LN/24---by the SB-ICP-MS---details first then abbreviate ---what is this ????

LN/33---bio-indicator for pollution or just for environmental  monitoring ?????

LN/39-40----add the year of the cited reference

LN/46-47----the cited references should be rearranged according to time date from the old ones to the new ones ---2003----2005---2007---2011---2020----etc

LN47---which part (tissue ) observed more accumulated than others ???

LN/50---add heavy metal instead of metallic

LN/51-52----the cited references also should rearranged according to alpha bet (for example : Hung et al .,1989 then Han et al .,1998 then Hung et al ., 1998

LN/57-58---rearrange according to date (1989---1990---1993--1995---1996

LN/62---salinity water needed for culturing ---clear that

LN/63-64---delete as it is repeated

LN/280-281---rearrange

Can you explain why some metals has a bioaccumulation in a certain layer more than the other layer (inner, middle , outer)???

The author did not mention any thing about the bad health hazards of the heavy metals for consumers---why ???

There is no conclusion

There is no ethical approval code or statement (should be )

There is no recommendations ---how we can avoid the health hazards of the polluted metals ???

LN/386-387---rearrange

Why you did not try to make a histological sections to the soft tissues or hard shell using decalcifier and staining with H&E stain to detect some bad impact of these metals ????

The most cited references need to be more update

Ref(3)---of a missing data (chapter or pages  ????)

Ref(7)---2017 delete (repeated)

Ref(10)----missing data

Ref(12&18)---old reference (update)

Ref(16&24)---contained more than 6 authors (should at the maximum 6 plus etal with the last ones )

Ref(28&29)---written in a different language (may be Chinese) same language should be

There is no gross pictures for the bivalves and the culturing 

Author Response

LN/34---add bivalves, Mollusks and filter feeding behavior to the keywords

Ans: LN/36---Already added in the keywords.

LN/24---by the SB-ICP-MS---details first then abbreviate ---what is this???

Ans: LN/25---Already added the detail of the Solution Based Inductively Coupled Plasma Mass Spectrometry (SB-ICPMS).

LN/33---bio-indicator for pollution or just for environmental monitoring?

Ans: LN/35---In this study we evaluate which elements in the shell of M. meretrix have potentials to be bio-indicators for assessing heavy metal in the environment for environmental monitoring.

LN/39-40----add the year of the cited reference

Ans: LN/579---Already revised the format of the reference.

LN47---which part (tissue) observed more accumulated than others???

Ans: LN66---In the muscle tissue observed more accumulation than others.

LN/50---add heavy metal instead of metallic

Ans: LN/69---Already revised to “heavy metal”.

LN/62---salinity water needed for culturing ---clear that

Ans: LN/80---Already revised the expression.

LN/63-64---delete as it is repeated

Ans: LN/80-81---Already deleted.

LN/280-281---rearrange

Ans: LN/579---Already revised the format of the reference.

Can you explain why some metals has a bioaccumulation in a certain layer more than the other layer (inner, middle, outer)???

Ans: Results in this study showed a different accumulation pattern of elements in the shall of M. meretrix, which may due to lots of reason. In previous study, different element accumulation rate may be impacted by ocean acidification, climate change, short-term accident (such as: gasoline spill, industries emission water with specific heavy metal elements) during the growing stage of bivalves, and the change of the accumulation rate of some specific elements will showed that a bioaccumulation in a certain layer more than the other layer rely on when those event happened during the growth stage of bivalves. And in this study, we were trying to find out the potential to observe how those event impact the bioaccumlation to the M. meretrix.

The author did not mention any thing about the bad health hazards of the heavy metals for consumers---why???

Ans: LN/501-502---In this research, we mainly focus on the potential of the using of the shall of hard clam M. meretrix as a bioindicator of heavy metal accumulation. So we did not discuss about the bad health hazard caused by those element to consumer at first. After revised, we have mentioned about the exceed of heavy metal may cause some bad hazard to human being and environment.

There is no conclusion

Ans: LN/545--- We had already added the conclusion part in the revised manuscript. ‘‘In conclusion, the hard clam M. meretrix, which is a wild distributed bivalve in the coast of Taiwan, its shell may have potential to be a good bio-indicator to show the environment change for the elements such as Na, Sr, Mg, B, Mn, P, Ba, Li and Cu. Moreover, since that Na and B showed an increasing accumulation along with the development and Sr, Fe and Mn with a decreasing trend along with growth, detecting for those elements should be handled more carefully. The correlation between the heavy metal concentration in water and sediment and its accumulation patern in the clam shell should be clarified by further studies.’’

There is no ethical approval code or statement (should be)

Ans: Already provided.

There are no recommendations ---how we can avoid the health hazards of the polluted metals???

Ans: LN/501-502---In this research, we mainly focus on the potential of the using of the shall of hard clam M. meretrix as a bioindicator of heavy metal accumulation. So we did not discuss about the bad health hazard caused by those element to consumer at first. After revised, we have mentioned about the exceed of heavy metal may cause some bad hazard to human.

LN/386-387---rearrange

Ans: LN/579---Already revised the format of the reference.

Why you did not try to make a histological sections to the soft tissues or hard shell using decalcifier and staining with H&E stain to detect some bad impact of these metals????

Ans: Thanks for the recommendation, we will undergo more detail research about the impact of those heavy metal in M. meretrix in the future. As in this study, we only focus on the research of the accumulation pattern if the shall if M. meretrix and did not do more physiological study about the impact of the heavy metal.

Problems about reference

LN/46-47----the cited references should be rearranged according to time date from the old ones to the new ones ---2003----2005---2007---2011---2020----etc

LN/51-52----the cited references also should rearranged according to alpha bet (for example : Hung et al .,1989 then Han et al .,1998 then Hung et al ., 1998

LN/57-58---rearrange according to date (1989---1990---1993--1995---1996

The most cited references need to be more update

Ref(3)---of a missing data (chapter or pages????)

Ref(7)---2017 delete (repeated)

Ref(10)----missing data

Ref(12&18)---old reference (update)

Ref(16&24)---contained more than 6 authors (should at the maximum 6 plus etal with the last ones )

Ref(28&29)---written in a different language (may be Chinese) same language should be

Ans: LN/579---Already revised the format of the reference.

There are no gross pictures for the bivalves and the culturing

Ans: LN/183---We had added the picture in the new manuscript.

Reviewer 2 Report

The manuscript prepared by Lin et al., entitled: “Evaluation of the potential heavy metals in the hard clam shell which can be used as the bio-indicator” presents results of the original research interesting from the aspect of application of Meretix meretix as bioindicator organism and detection of exposure to heavy metals in aquatic ecosystems.

The present study has at least one great limitation.

1. The authors provide results on levels of heavy metals in shells of the M. meretix, with the aim to point out to those that highly accumulate in the shells and that could therefore be used as biomarkers of exposure. However, to come to such conclusion, concentrations of heavy metals in water at sites and aquatic farms from which M. meretix specimens were collected should be measured, and correlated to concentrations in the shells. This is very important issue that has to be addressed to fulfill the present aim of the study. If it is not possible to provide results on concentrations of heavy metals at these sites, then authors should put the effort to provide adequate literature data from similar sites, information on expected concentrations, and discuss the results also in this context. To overcome this limitation, an option would also be to present the results as they are, as information on level of heavy metals in M. meretix as biomarkers of exposure, but without intention to search for the most sensitive biomarkers of exposure. Such approach, however, would reduce the overall scientific quality of the manuscript.

There are also some other major concerns related to overall quality of the manuscript that have to be addressed.

1. The  term “bioindicator” is being constantly misused throughout the manuscript, and also in the title. The term bioindicator is used for living organisms (animals, plants, microorganisms...), and in the context of this manuscript it is M. meretix. Heavy metals, in the context of this study, can not be called bioindicators, but they can be considered as biomarkers of exposure.

In parts of the text where bioindicator remains as appropriate term (when it refers to organisms), it should be written without hyphen (bioindicator instead of bio-indicator).

2. Clear and comprehensive conclusion should be added as a separate section of the manuscript, after Discussion.

3. The entire manuscript requires thorough English language editing.

Author Response

The manuscript prepared by Lin et al., entitled: “Evaluation of the potential heavy metals in the hard clam shell which can be used as the bio-indicator” presents results of the original research interesting from the aspect of application of Meretix meretix as bioindicator organism and detection of exposure to heavy metals in aquatic ecosystems.

The present study has at least one great limitation.

  1. The authors provide results on levels of heavy metals in shells of the meretix, with the aim to point out to those that highly accumulate in the shells and that could therefore be used as biomarkers of exposure. However, to come to such conclusion, concentrations of heavy metals in water at sites and aquatic farms from which M. meretix specimens were collected should be measured, and correlated to concentrations in the shells. This is very important issue that has to be addressed to fulfill the present aim of the study. If it is not possible to provide results on concentrations of heavy metals at these sites, then authors should put the effort to provide adequate literature data from similar sites, information on expected concentrations, and discuss the results also in this context. To overcome this limitation, an option would also be to present the results as they are, as information on level of heavy metals in M. meretix as biomarkers of exposure, but without intention to search for the most sensitive biomarkers of exposure. Such approach, however, would reduce the overall scientific quality of the manuscript.

Ans: Thank reviewer for the recommendation, we knew that the lacking of the data of heavy metals concentration in water at sites and aquatic farms may hinder the accuracy of the approach that M. meretrix can be a bioindicator to the heavy metal exposure. Since in this study, we mainly focus on finding out the potential of the ability for using M. meretrix as the bioindicator. We had found that there were different accumulation patterns between some elements (Na, B, Sr, Fe and Mn) along their growth stage, and also others elements show significantly differ in inner, middle and outer. According to the results, we will conduct a more detail research about water/sediment and the correlation between those data and the accumulation pattern we found in this study.

There are also some other major concerns related to overall quality of the manuscript that have to be addressed.

  1. The term “bioindicator” is being constantly misused throughout the manuscript, and also in the title. The term bioindicator is used for living organisms (animals, plants, microorganisms...), and in the context of this manuscript it is M. meretix. Heavy metals, in the context of this study, can not be called bioindicators, but they can be considered as biomarkers of exposure.

In parts of the text where bioindicator remains as appropriate term (when it refers to organisms), it should be written without hyphen (bioindicator instead of bio-indicator).

Ans: Thank for the recommendation, we had corrected the “bio-indicator” to “bioindicator” in our latest manuscript. As for the misusing of bioindicator and biomarker, we had also corrected it in it.

  1. Clear and comprehensive conclusion should be added as a separate section of the manuscript, after Discussion.

Ans: LN/545--- We had already added the conclusion part in the revised manuscript. ‘‘In conclusion, the hard clam M. meretrix, which is a wild distributed bivalve in the coast of Taiwan, its shell may have potential to be a good bio-indicator to show the environment change for the elements such as Na, Sr, Mg, B, Mn, P, Ba, Li and Cu. Moreover, since that Na and B showed an increasing accumulation along with the development and Sr, Fe and Mn with a decreasing trend along with growth, detecting for those elements should be handled more carefully. The correlation between the heavy metal concentration in water and sediment and its accumulation patern in the clam shell should be clarified by further studies.’’

  1. The entire manuscript requires thorough English language editing.

Ans: We had already revised the typographical errors and grammar mistake in the manuscript. 

Round 2

Reviewer 2 Report

The revised version of the manuscript prepared by Lin et al., now entitled: “Evaluation the potential of the hard clam (Meretrix meretrix) shell which can be used as the bioindicator for heavy metal accumulation has undergone some changes suggested in the first-round review. However, they have not been conducted systematically, precisely and thoroughly, but are rather superficial and sporadic.

GENERAL COMMENTS

1. Although authors, in their response to the reviewer, state that they corrected “bio-indicator” to “bioindicator”, this correction has not been done throughout the manuscript (for example lines 22, 25, 47, 83, 303, 345, 433, 436).

 2. Although authors, in their response to the reviewer, state that the term “bioindicator” is being properly used in the revised version of the manuscript, there are still many examples of its misusage (as explained in the first-round review), such as in lines 25, 34, 83, 303…

3. Although authors, in their response to the reviewer, state that they revised the typographical errors and grammar mistake in the manuscript, actually only a few changes have been made related to English editing and the entire manuscript still requires serious changes related to the quality of language, scientific expressions and clarity of sentences in general. The title of the manuscript also requires language editing.

OTHER COMMENTS:

1.     Different methods are stated in the abstract and Material and methods (lines 26 and 106, SB-ICPMS) and Introduction (line 80, LA-ICPMS).

2.     The aim of the study is not clearly written.

3.     Lines 101-102: why was the DNA species identification done for cultured specimens? The authors do not mention this later in the text.

Author Response

Response to Reviewer 2

LN3, 22, 25, 34,36,103, 140,168,474, 540,881,883, 968,971, 973, 974, 975

For the misused of the term “bioindicator” and “biomarker” we had revised all the mistake in the latest article.

The entire manuscript requires thorough English language editing.

Ans: We have gone through a grammar editing in our latest manuscript.